# Computer Simulation of Coke Sediments Burning from the Whole Cylindrical Catalyst Grain

**Olga S. Yazovtseva** [1,*]#, **Irek M. Gubaydullin** [2], **Elizaveta E. Peskova** [1], **Lev A. Sukharev** [1]
**and Andrey N. Zagoruiko** [3]

[1] Faculty of Mathematics and Information Technologies, National Research Mordovia State University,
68 Bolshevistskaya Str., Saransk 430005, Russia

[2] Institute of Petrochemistry and Catalysis of the Russian Academy of Sciences, Ufa 450075, Russia

[3] Boreskov Institute of Catalysis SB RAS, Novosibirsk 630090, Russia

[*] Correspondence: kurinaos@gmail.com; Tel.: +7-927-974-8917

**Abstract:** The article is devoted to the development of the mathematical model of oxidative regeneration of the cylindrical catalyst grain. The model is constructed using a diffusion approach to modeling catalytic processes. The model is based on the equations of material and thermal balance. Mass transfer in the catalyst grain is carried out due to diffusion and the Stefan flow resulting from a decrease in the reaction volume during sorption processes. Chemical transformations of substances are taken into account as a source term in the equation. The thermal balance of the catalyst grain is described by a thermal conductivity equation, with an inhomogeneous term responsible for heating the grain during exothermic chemical reactions. The effective coefficients of heat capacity and thermal conductivity of the catalyst grain, which are determined taking into account the porosity of the grain depending on temperature, were used to calculate the thermal balance of the catalyst grain. The dependencies are approximated using the method of least squares based on experimental data. Different boundary conditions for the developed model allow calculating the main characteristics of the oxidative regeneration process for a whole catalyst grain under different conditions. The mathematical model of oxidative regeneration of a cylindrical catalyst grain is described by a stiff system of differential equations. Splitting by physical processes is applied to avoid computational difficulties. The calculation of flows is carried out sequentially: first, chemical problems are solved using the Radau method, then the diffusion and thermal conductivity equations are solved by the finite volume method. The result of the algorithm implemented in C++ is a picture of the distribution of substances and temperature along the cylindrical grain of the catalyst.

**Keywords:** oxidative regeneration; chemical kinetics; diffusion equation; thermal conductivity equation; splitting by physical processes; numerical methods; mathematical modeling

**MSC:** 35Q92

## 1. Introduction

The development of mathematical models of chemical and technological processes has been an actual problem for many decades. Mathematical modeling allows to avoid full-scale experiments and has proven itself as an effective method for solving many technological problems [1,2].

An important stage of industrial oil refining is the process of cleaning the catalyst from the coke sediments that accumulate on the surface of the layer and inside the catalyst grains [3]. One of the least expensive cleaning methods is oxidative regeneration—burning of coke sediments with oxygen-containing gas. An actual problem is to ensure such conditions for the flow of oxidative regeneration, in which coke burns out quickly enough, but at the same time there is no overheating of the catalyst grain.

Currently, the oil and gas industry is focused on the maximum selectivity of raw materials. This is often reached by using complex technological schemes of reactors with significant gas-dynamic resistance. The problem of gas-dynamic resistance can be solved by developing catalysts of various geometric shapes and sizes. The variety of catalyst types leads to the need to develop classes of mathematical models that allow calculating the main characteristics of the oxidative regeneration process for various grain forms, sediment compositions and catalyst materials.

The article [4] presents a study of processes inside a spherical catalyst grain in the presence of coke sediments. It is noted that coke in the pores of the catalyst significantly reduces diffusion. It is also noted that diffusion models are quite complex; in most cases, they require a numerical solution.

The article [5] is devoted to the study of the influence of coke on heterogeneous catalysis. As in [4], a strong influence of coke in the catalyst pores on internal diffusion was noted. The Monte Carlo method was used to simulate the deposition of coke in the pores. Similarly, the removal of coke from the pores is modeled. This shows that coke can be removed unevenly over time.

The paper [6] presents a two-dimensional (bed length and pellet radius coordinates), two-temperature (gas and catalyst phases) catalyst bed model used for simulation of the oxidative regeneration of coked hydrofluorination CrF3/MgF2 catalyst in the adiabatic reactor. The study discusses the rather complicated influence of process parameters on the main characteristics of the regeneration procedure. In particular, it was shown that feeding of the regenerating gas with temperatures lower than the coke ignition point may lead to the formation of a superadiabatic combustion wave with extremely high temperatures, even under low inlet oxygen concentration.

The article [7] presents a model of adsorption–catalytic removal of volatile organic compounds. This model also gives the successful example of the adiabatic catalyst bed model, where the interplay of external and internal mass transfer limitations with exothermic reaction may also give the rather complicated and nontrivial behavior of the oxidative regeneration process.

These models are rather detailed, but they all consider the catalyst pellets as spheres or propose the reduction of other shapes to pseudospheres. Such an approach may seriously worsen the simulation adequacy and accuracy in cases of nonspherical particles (cylinders, rings, multilobes, etc.).

The article [8] contains a coked naphtha reforming catalyst (Pt/Re-Al2O3) regeneration model. Modeling is carried out at the level of individual particles and a fixed catalyst bed, the rate of coke burnout is investigated with varying some characteristics of the catalyst. This work is related to cylinder-shaped pellets, though using the simplified model assuming the absence of axial coke and oxygen gradients in the pellet, which is correct only for cylinders with some fixed diameter and infinite length; this model is not applicable to real shapes, where the diameter is commensurable with length.

Therefore, the correct simulation of catalyst pellets with nonsphere shapes is an important problem, definitely lacking the corresponding research, especially in the case of transient processes such as oxidative regeneration.

A common problem that arises in models that combine heat and mass transfer, chemical reactions and diffusion is the complexity of systems of equations describing these models. Well-known computational methods and algorithms are often ineffective for solving such problems and do not allow numerous computational experiments to be carried out quickly. A possible solution seems to be the reduction of complex nonlinear models to simpler ones that reflect some of the main characteristics [9].

The examples of such an approach in modeling the oxidation of organic substances is the model of an unresponsive core and the model of progressive transformation.

The unreacted core model is used for modeling of nonporous environments. If the catalyst grain is nonporous, then reactions can only take place on its surface, which leads to a significant simplification of the model [10]. This model is used for layer-by-layer

burnout of the reagent with a decrease in the reaction zone. It takes into account kinetics and diffusion. However, many researchers note that adequate modeling of some processes requires consideration of flows [11]. Moreover, this model is characterized by the presence of a movable boundary, which adequately describes the process of particle reduction over time [12] but does not correspond to the process of oxidative regeneration.

The progressive conversion model can be used in the case of a high level of porosity of the catalyst grain [13]. This model adequately describes the processes accompanied by rapid diffusion of substances into the grain, however, it does not take into account the change in the reaction surface area that inevitably occurs during coke burning. Moreover, a necessary condition for the application is the absence of the influence of solid concentrations on the chemical reactions rate, which is impossible in the presence of sorption processes, as well as the assumption of quasi-stationarity of the process [14].

Previously, the authors obtained a mathematical model for burning coke sediments from a catalyst spherical grain [15,16], developed on the basis of the well-known model [17,18]. It is described by a system of parabolic equations corresponding to the thermal, diffusion and chemical processes accompanying oxidative regeneration.

The present model uses a similar multistage kinetic scheme that takes into account the gas and solid phases of the reaction. Regeneration of the catalyst based on $Al_2O_3$ is considered. Kinetic characteristics are taken from the work [17].

Effective coefficients of diffusion, thermal conductivity and heat capacity, depending on temperature, are used to calculate the main characteristics of the regeneration process. The dependencies of heat capacity and thermal conductivity are approximated by polynomials based on experimental data, and the diffusion coefficient is obtained based on the Knudsen diffusion coefficient.

The grain geometry makes its own adjustments to the dynamics of the main processes' characteristics, despite the uniform nature of the processes occurring in spherical and cylindrical grains. The material balance equations are supplemented with a convective term corresponding to mass transfer along the axis of the cylinder to account for the processes in a cylindrical grain. Similar changes have been made to thermal conductivity—it is necessary to take into account its propagation along the cylindrical axis in addition to the propagation of heat along the radius.

The calculation of the Stefan flow velocity has been corrected due to the complexity of the diffusion processes description, which in turn leads to the complexity of its discrete analogue. The flow in the spatial cell with the number $[i, j]$ is integrated using the values of grain concentrations and temperature in the cell $[i - 1, j - 1]$, despite that only the first-order derivative is used in the model. In addition, the transition from a one-dimensional problem in the case of central symmetry to a two-dimensional axisymmetric problem leads to a significant increase in computational complexity for refinement of the spatial grid. All this leads to the use of parallel computing as an effective means of reducing time costs.

The aim of this work is to construct a mathematical model of the oxidative regeneration of solid cylindrical catalyst grains, as well as to develop and implement an effective numerical algorithm for calculating concentrations of substances involved in chemical reactions and the temperature of the catalyst grain.

## 2. Development of a Mathematical Model of Processes in a Cylindrical Catalyst Grain

The most important part of the mathematical description of any process involving chemical reactions is their kinetic model. This paper presents a description of the burning of coke sediments, presented in the works [17–20], as a kinetic scheme:

$$2\theta_C + O_2 \longrightarrow 2\theta_{CO}, \quad W_1 = k_1(T_z)\,\theta_3^2\,y_1;$$
$$\theta_{CO} + O_2 \longrightarrow \theta_{CO} + CO_2, \quad W_2 = k_2(T_z)\,\theta_2\,y_1;$$
$$\theta_{CO} \longrightarrow \theta_C + CO, \quad W_3 = k_3(T_z)\,\theta_2;$$
$$\theta_{CH_2} + O_2 \longrightarrow \theta_{CO} + H_2O, \quad W_4 = k_4(T_z)\,\theta_1\,y_1; \tag{1}$$
$$\theta_{CO} + \theta_{CO} \longrightarrow 2\theta_C + CO_2, \quad W_5 = k_5(T_z)\,\theta_2^2;$$
$$\theta_{CH_2} \rightleftharpoons \theta_C + Z_{H_2}, \quad W_6 = k_6(T_z)\,\frac{\rho_C}{R_C}\,(\theta_1^* - z_1);$$
$$\theta_{CO} \rightleftharpoons \theta_C + Z_O, \quad W_7 = k_7(T_z)\,\frac{\rho_C}{R_C}\,(\theta_2^* - z_2).$$

Here, $W_i$, $i = \overline{1,7}$—rates of chemical interaction stages, dimension $W_r$, $r = \overline{1,5}$—mole/(m$^2$·s), $W_6$ and $W_7$—g/(m$^2$·c); $T_z$—catalyst grain temperature, K; $k_j(T_z)$, $j = \overline{1,7}$—rate constants of chemical interaction stages, the dimension of $k_j$ coincides with the dimension of $W_j$; $\theta_l$, $l = \overline{1,3}$—the degree of covering of the coke surface with various carbon complexes ($\theta_1$—hydrogen-carbon complex, $\theta_2$—oxygen–carbon complex and $\theta_3$—free carbon surface); $y_1$—oxygen concentration in the gas phase in molar fractions; $z_1$ and $z_2$—concentrations of hydrogen and oxygen in the coke layer in mass fractions; $\theta_1^*$ and $\theta_2^*$—the amount of hydrogen and oxygen adsorbed by coke relative to the current state of the surface of coke sediments; $\rho_C$ and $R_C$ are the density (g/m$^3$) and the average radius of the pellets (m) of coke. Meanwhile,

$$\theta_1 + \theta_2 + \theta_3 = 1. \tag{2}$$

The first reaction in the scheme (1) is a multistage dissociative adsorption of oxygen, leading to the formation of an oxygen–carbon complex on the surface of the catalyst grain. The second stage of the scheme (1) is the oxidation of the oxygen–carbon complex with the release of carbon dioxide. The third stage describes the destruction of the carbon–oxygen complex with the release of a free carbon surface and the release of carbon monoxide [19]. The fourth reaction describes the multistage oxidation of a hydrogen–carbon complex with the formation of an oxygen–carbon complex and water vapor. The fifth stage corresponds to the recombination of the oxygen–carbon complex with the release of a free carbon surface. The concomitant release of carbon dioxide is the result of oxygen desorption, which, without creating special conditions, is possible only with reaction products [21].

The determination of the rates of the first five stages of oxidative regeneration is subject to the law of conservation of active masses. This assumption allows us to obtain expressions for reaction rates consistent with experimental data.

The sixth and seventh reactions in the (1) scheme are heterogeneous processes that lead to the release of the free carbon surface of the coke pellet. Taking into account the density and radius is necessary due to the fact that the coke structure, and hence its behavior in the oxidation reaction, depend on the density and size of the coke granules. The expressions for the reaction rates $W_6$ and $W_7$ are derived from empirical data. The reaction scheme and its detailed description are given in [17].

The constants $k_i^{\mathrm{op}}$ are given in the works [17] as the constants of reaction rates at temperature $T_{\mathrm{op}} = 793$ K, $i = \overline{1,7}$. In this case, it is necessary to take into account the change in the rate constants of reactions with temperature changes, expressed from the Arrhenius equation:

$$k_i(\Theta) = k_i^{\mathrm{op}} \exp\left(\frac{E_i}{R\,T_z}\left(1 - \frac{1}{\Theta}\right)\right), \tag{3}$$

Here, $\Theta = \dfrac{T_z}{T_{\mathrm{op}}}$—dimensionless catalyst grain temperature obtained by dividing the current temperature by $T_{\mathrm{op}}$; $E_i$—activation energies, J/mole; $R = 8.31$ J/mole/K.



It should be noted that, as coke sediments are burned during oxidative regeneration, the radius and area of coke pellets decrease. The following dependencies are given in [17]:

$$R_C = R_C^0 \left( \frac{q_c}{q_c^0} \right)^{\frac{1}{3}}, \quad S_k = S_k^0 \left( \frac{q_c}{q_c^0} \right)^{\frac{2}{3}}, \tag{4}$$

where $q_C$ and $q_C^0$ are the current and initial mass fraction of coke sediments on the catalyst grain, respectively; $R_C^0$ is the initial average radius of coke granules on the catalyst grain, m; and $S_k^0 = \frac{3q_c^0}{R_C^0 \rho_C}$—the initial specific surface area of coke sediments, which plays the role of the reaction surface in the burning process, $m^2/g$.

The flow of catalytic processes in the catalyst grain is classically described by the equations of thermal and material balance. The developed model is called diffusion, and it is based on the laws of conservation of mass and energy.

The reduction of the three-dimensional formulation of the problem to its two-dimensional analogue, the axisymmetric problem, has proven itself well. This approach assumes a transition from a three-dimensional Cartesian rectangular coordinate system to a cylindrical coordinate system with two spatial variables: $r$ corresponds to a cylindrical radius, and $z$ corresponds to the semiaxis of the cylinder.

The thermal balance implies taking into account the distribution of heat through the catalyst grain and the thermal effect of the reaction. It can be described using the thermal conductivity equation, while the role of an inhomogeneous term in its composition is played by the heating of the grain due to the release of heat as a result of exothermic reactions. The equation of thermal conductivity will have the form:

$$c^* \frac{\partial T_z}{\partial t} = \frac{1}{r} \frac{\partial}{\partial r} \left( r \lambda^* \frac{\partial T_z}{\partial r} \right) + \lambda^* \frac{\partial^2 T_z}{\partial z^2} + \gamma_k S_k \sum_{j=1}^{5} Q_j W_j, \tag{5}$$

where $c^*$ is the effective heat capacity coefficient of the catalyst, $J/m^3/K$; $\lambda^*$—effective thermal conductivity coefficient of the catalyst, $W/m/K$; $\gamma_k$—bulk density of the catalyst, $g/m^3$; and $Q_j, j = \overline{1,5}$—thermal effect of the $j$-th reaction stage in the (1) scheme.

The effective heat capacity coefficient calculated by the formula [22] is taken as $c^* = c^*(T_z)$:

$$c^*(T_z) = (1 - \varepsilon)(\rho_k (A_k T_z^2 + B_k T_z + C_k)(1 - q_c) + (A_C T_z^2 + B_C T_z + C_C) q_c), \tag{6}$$

where $\rho_k$ is the density of the catalyst material; $A_k, B_k, C_k$ and $A_C, B_C, C_C$ are the coefficients of the polynomial temperature dependence of the specific heat capacity of the catalyst material and coke, respectively; and $\varepsilon$ is the porosity of the catalyst grain.

The effective coefficient of thermal conductivity $\lambda^*$ is calculated by the formula [23]:

$$\lambda^*(T_z) = (1 - \varepsilon)(A_L T_z^2 + B_L T_z + C_L), \tag{7}$$

where $A_L, B_L, C_L$ are the coefficients of the polynomial temperature dependence of the thermal conductivity of the catalyst material.

The source term $\gamma_k S_k \sum_{j=1}^{5} Q_j W_j$ for the thermal conductivity equation is derived from accounting for chemical transformations on grain and takes into account its heating due to the exothermic nature of reactions in the (1) scheme.

The material balance of the catalyst grain is described on the basis of Fick's law:

$$\varepsilon \frac{\partial y_i}{\partial t} = \frac{1}{r} \frac{\partial}{\partial r} \left( r D^* \frac{\partial y_i}{\partial r} - r \mu y_i \right) + \frac{\partial}{\partial z} \left( D^* \frac{\partial y_i}{\partial z} - \mu y_i \right) + \frac{\gamma_k S_k}{c_0} \sum_{j=1}^{7} \nu_{ij} W_j, \tag{8}$$

where $y_i$, $i = \overline{1,4}$—concentrations of substances in grain pores in molar fractions; $D^*$—effective diffusion coefficient, m$^2$/s; $\mu$—the velocity of the Stefan flow, m/s; $c_0$—molar gas density, mole/m$^3$; $\nu_{ij}$, $i = \overline{1,4}$, $j = \overline{1,7}$—stoichiometric coefficients of substances.

The detailed kinetic scheme assumes heterogeneous reactions occurring with a change in the reaction volume, which leads to the loss of some volume of the reaction gas phase. It follows from the law of mass conservation that the Stefan flow plays an important role in the regeneration process in addition to the diffusion flow. The calculation of its velocity is implemented by summing the equations of the material balance and equating to zero the changes in the sum of the substance concentrations of the gas phase in time and space. It is necessary to take into account two components of the Stefan flow—along the cylindrical radius and along the axis of the cylinder—since the assumption of a uniform grain structure is accepted in the work. The velocity of the Stefan flow does not depend on time, but it depends on space and is a scalar quantity:

$$\frac{1}{r}\frac{\partial}{\partial r}(r\mu) + \frac{\partial \mu}{\partial z} = \frac{\gamma_k S_k}{c_0}(W_1 + W_3 + W_5). \tag{9}$$

It is necessary to take into account the consumption and formation of substances due to chemical reactions in addition to the diffusion and Stefan flow—the source term in the diffusion equation.

The material balance also needs to be supplemented with equations describing changes in the substances' concentrations in the solid phase of the reaction:

$$\begin{cases} \dfrac{dq_c}{dt} = -M_C S_k(W_2 + W_3 + W_5), \\ \dfrac{dz_1}{dt} = \dfrac{S_k}{q_C}(W_6 + z_1 M_C(W_2 + W_3 + W_5)), \\ \dfrac{dz_2}{dt} = \dfrac{S_k}{q_C}(W_7 + z_2 M_C(W_2 + W_3 + W_5)), \\ \dfrac{d\theta_1}{dt} = -\dfrac{\gamma_k S_k}{c_0} W_4 - S_k W_6, \\ \dfrac{d\theta_2}{dt} = \dfrac{\gamma_k S_k}{c_0}(2W_1 - W_3 + W_4 - 2W_5) - S_k W_7, \end{cases} \tag{10}$$

where $M_C$ is the molar mass of coke, g/mole.

Various types of cylindrical catalysts are used in industrial processes, which can be conditionally divided into two large groups: whole and hollow cylinders. The processes occurring in whole and hollow cylinders are the same, however, in terms of mathematical physics, boundary conditions for the mathematical model play an important role, implying for this task the heat and mass transfer of grain with the environment.

In the case of a whole cylindrical catalyst grain, heat and mass transfer occurs through the cylindrical wall and the base of the cylinder. Initial boundary conditions have the form:

$$t = 0: \; q_c(0) = q_C^0, \; z_1(0) = z_1^0, \; z_2(0) = 0, \; \theta_1(0) = \theta_1^0, \; \theta_2(0) = 0, \; T_z(0) = T(0),$$

$$y_1(0) = y_1^0, \; y_i(0) = 0, i = \overline{2,4}, \; \mu = 0, \; \theta_2(0) = 0, \; T_z(0) = T(0),$$

$$r = 0, \; z = 0: \; \frac{\partial y_i}{\partial r} = \frac{\partial y_i}{\partial z} = 0, \; \frac{\partial T_z}{\partial r} = \frac{\partial \Theta}{\partial z} = 0; \tag{11}$$

$$r = R_z: \lambda^* \frac{\partial T_z}{\partial r} = \alpha(T(0) - T_z), \; \frac{\partial y_i}{\partial r} = 0, i = \overline{1,4},$$

$$z = L: \lambda^* \frac{\partial T_z}{\partial z} = \alpha(T(0) - T_z), \; \frac{\partial y_i}{\partial z} = 0, i = \overline{1,4}.$$

Thus, the mathematical model of oxidative regeneration of the cylindrical catalyst grain has the form:

$$
\begin{cases}
c^* \dfrac{\partial T_z}{\partial t} = \dfrac{1}{r}\dfrac{\partial}{\partial r}\left(r\lambda^* \dfrac{\partial T_z}{\partial r}\right) + \lambda^* \dfrac{\partial^2 T_z}{\partial z^2} + \gamma_k S_k \sum\limits_{j=1}^{5} Q_j W_j, \\[2mm]
\varepsilon_k \dfrac{\partial y_i}{\partial t} = \dfrac{1}{r}\dfrac{\partial}{\partial r}\left(rD^*\dfrac{\partial y_i}{\partial r} - r\mu y_i\right) + \dfrac{\partial}{\partial z}\left(D^*\dfrac{\partial y_i}{\partial z} - \mu y_i\right) + \dfrac{\gamma_k S_k}{c_0}\sum\limits_{j=1}^{5} v_{ij} W_j, \\[2mm]
\dfrac{1}{r}\dfrac{\partial}{\partial r}(r\mu) + \dfrac{\partial \mu}{\partial z} = \dfrac{\gamma_k S_k}{c_0}(W_1 + W_3 + W_5). \\[2mm]
\dfrac{\partial q_c}{\partial t} = -M_C S_k (W_2 + W_3 + W_5), \\[2mm]
\dfrac{\partial z_1}{\partial t} = \dfrac{S_k}{q_C}(W_6 + z_1 M_C (W_2 + W_3 + W_5)), \\[2mm]
\dfrac{\partial z_2}{\partial t} = \dfrac{S_k}{q_C}(W_7 + z_2 M_C (W_2 + W_3 + W_5)), \\[2mm]
\dfrac{\partial \theta_1}{\partial t} = -\dfrac{\gamma_k S_k}{c_0} W_4 - S_k W_6, \\[2mm]
\dfrac{\partial \theta_2}{\partial t} = \dfrac{\gamma_k S_k}{c_0}(2W_1 - W_3 + W_4 - 2W_5) - S_k W_7,
\end{cases}
\tag{12}
$$

with initial boundary conditions (11).

## 3. Dimensionless Model of Oxidative Regeneration of Cylindrical Grain

Taking into account flow and chemical phenomena within the framework of one mathematical model inevitably leads to computational difficulties [24]. This is due to rapid chemical reactions and a relatively low diffusion flow rate. There is the method of dimensionalization proposed by the authors in [15,16] used to reduce the multiscale of such processes.

The dimensionless model of coke burning for a cylindrical catalyst grain will take the form:

$$
\begin{cases}
\dfrac{\partial y_i}{\partial \tau} = \dfrac{1}{\varphi \varepsilon_k}\dfrac{1}{\rho}\dfrac{\partial}{\partial \rho}\left(\rho \dfrac{\partial y_i}{\partial \rho} - \rho \hat{\mu} y_i\right) + \dfrac{R_z}{\varphi \varepsilon_k L^2}\dfrac{\partial}{\partial l}\left(R_z \dfrac{\partial y_i}{\partial l} - L\hat{\mu} y_i\right) + \dfrac{\hat{S}}{\varepsilon_k}\sum\limits_{j=1}^{5} v_{ij}\omega_j, \\[2mm]
\dfrac{\partial \Theta}{\partial \tau} = \dfrac{\lambda^* \tau_k}{c^* R_z^2}\dfrac{1}{\rho}\dfrac{\partial}{\partial \rho}\left(\rho \dfrac{\partial \Theta}{\partial \rho}\right) + \dfrac{\lambda^* \tau_k}{c^* L^2}\dfrac{\partial \Theta}{\partial l} + \dfrac{\hat{S}c_0}{T_{op}c^*}\sum\limits_{j=1}^{5} Q_j\omega_j, \\[2mm]
\dfrac{1}{\rho}\dfrac{\partial}{\partial \rho}(\rho \hat{\mu}) + \dfrac{R}{L}\dfrac{\partial \hat{\mu}}{\partial l} = \varphi \hat{S}(-\omega_1 + \omega_3 + \omega_5), \\[2mm]
\dfrac{\partial q_c}{\partial \tau} = -\dfrac{M_C c_0}{\gamma_k}\hat{S}(\omega_2 + \omega_3 + \omega_5), \\[2mm]
\dfrac{\partial z_1}{\partial \tau} = \dfrac{c_0}{\gamma_k q_c}\hat{S}(\omega_6 + z_1 M_C(\omega_2 + \omega_3 + \omega_5)), \\[2mm]
\dfrac{\partial z_2}{\partial \tau} = \dfrac{c_0}{\gamma_k q_c}\hat{S}(\omega_7 + z_2 M_C(\omega_2 + \omega_3 + \omega_5)), \\[2mm]
\dfrac{\partial \theta_1}{\partial \tau} = -\hat{S}\left(\omega_4 + \dfrac{c_0}{\gamma_k}\omega_6\right), \\[2mm]
\dfrac{\partial \theta_2}{\partial \tau} = \hat{S}\left(2\omega_1 - \omega_3 + \omega_4 - 2\omega_5 - \dfrac{c_0}{\gamma_k}\omega_7\right).
\end{cases}
\tag{13}
$$

The dimensionless initial boundary conditions for a whole cylinder have the form:

$$
\tau = 0: \quad q_c(0) = q_C^0, \; z_1(0) = z_1^0, \; z_2(0) = 0, \; \theta_1(0) = \theta_1^0, \; \theta_2(0) = 0, \; \Theta(0) = \dfrac{T(0)}{T_{op}},
$$

$$
y_1(0) = y_1^0, \; y_i(0) = 0, i = \overline{2,4}, \; \hat{\mu} = 0; \; \rho = 0, \; l = 0: \quad \dfrac{\partial y_i}{\partial \rho} = \dfrac{\partial y_i}{\partial l} = 0, \; \dfrac{\partial \Theta}{\partial \rho} = \dfrac{\partial \Theta}{\partial l} = 0;
$$

$$\rho = 1: \quad \frac{\partial \Theta}{\partial \rho} = \frac{R_z \alpha}{\lambda^*}\left(\frac{T(0)}{T_{op}} - \Theta\right), \frac{\partial y_i}{\partial \rho} = 0, i = \overline{1,4}; \tag{14}$$

$$l = 1: \quad \frac{\partial \Theta}{\partial l} = \frac{L\alpha}{\lambda^*}\left(\frac{T(0)}{T_{op}} - \Theta\right), \frac{\partial y_i}{\partial l} = 0, i = \overline{1,4}.$$

## 4. The Computational Algorithm

### 4.1. The Difference Scheme for the Model of Oxidative Regeneration of Cylindrical Grain

Significant differences in the times of chemical reactions and transfer processes lead to difficulties in the development of a computational algorithm. When the model is of dimensionless form, the ratio of characteristic times decreases by several orders, but the system of equations remains stiff. The use of implicit methods does not give a gain in calculation time due to the complex right-hand side of the system for this system; therefore, the splitting method for physical processes is used for numerical modeling, which has proven itself well for calculating the dynamics of processes of various nature in a single model.

The equations of chemical kinetics are solved in a separate block by a three-stage method of the fifth order of Runge–Kutta accuracy, known as the Radau method, since the main computational difficulties are associated with fast chemical reactions [25].

The finite volume method is applied to solve the diffusion and thermal conductivity equations. It has the first order of approximation in time and the second order in space. The calculation error can be reduced by grinding the spatial grid.

The difference analogue for the system of balance equations of the (13) model has the form:

$$
\begin{cases}
\Theta_{i,j}^{n+1} = \Theta_{i,j}^n + \dfrac{\Delta t\, \lambda^*}{D^* \varphi i h_r^2}\left((i+0.5)\left(\Theta_{i+1,j}^n - \Theta_{i,j}^n\right) - (i-0.5)\left(\Theta_{i,j}^n - \Theta_{i-1,j}^n\right)\right) + \\
\qquad\qquad + \dfrac{\Delta t\, \lambda^* \tau_k}{c^* L^2} \dfrac{\Theta_{i,j+1}^n - 2\Theta_{i,j}^n + \Theta_{i,j-1}^n}{h_z^2} + \left(\dfrac{\hat{S} c_0}{T_{op} c^*}\sum_{o=1}^{5} Q_o \omega_o\right)^n_{i,j}, \\[4pt]
y_{i,j}^{n+1} = y_{i,j}^n * + \dfrac{\Delta t}{\varepsilon_k \varphi i h_r^2}\left((i+0.5)\left(y_{i+1,j}^n - y_{i,j}^n\right) - (i-0.5)\left(y_{i,j}^n - y_{i-1,j}^n\right)\right) - \\
\qquad - \dfrac{\Delta t}{\varepsilon_k \varphi i h_r^2}\left((i+0.5)\left(\hat{\mu}_{i+1,j}^n y_{i+1,j}^n - \hat{\mu}_{i,j}^n y_{i,j}^n\right) - (i-0.5)\left(\hat{\mu}_{i,j}^n y_{i,j}^n - \hat{\mu}_{i-1,j}^n y_{i-1,j}^n\right)\right) + \\
\qquad + \dfrac{\Delta t\, R_z^2}{\varepsilon_k \varphi L h_z^2}\left((j+0.5)\left(y_{i,j+1}^n - y_{i,j}^n\right) - (j-0.5)\left(y_{i,j}^n - y_{i,j-1}^n\right)\right) - \\
\qquad - \dfrac{\Delta t}{\varepsilon_k \varphi i h_z^2}\left((j+0.5)\left(\hat{\mu}_{i,j+1}^n y_{i,j+1}^n - \hat{\mu}_{i,j}^n y_{i,j}^n\right) - (j-0.5)\left(\hat{\mu}_{i,j}^n y_{i,j}^n - \hat{\mu}_{i,j-1}^n y_{i,j-1}^n\right)\right) + \\
\qquad\qquad\qquad\qquad\qquad + \left(\dfrac{\hat{S}}{\varepsilon_k}\sum_{o=1}^{5} v_o \omega_o\right)^n_{i,j}, \\[4pt]
\hat{\mu}_{i,j}^n = \dfrac{1}{h_r + h_z}\left(h_r \hat{\mu}_{i,j-1}^n - \dfrac{h_z}{i-1}\hat{\mu}_{i-1,j}^n + h_r \hat{\mu}_{i-1,j}^n + h_z h_r\left(\varphi \hat{S} \sum_{o=1}^{5} v_o \omega_o\right)^n_{i-1,j-1}\right).
\end{cases}
\tag{15}
$$

The dimensionalization reduces the integration area to the square $[0,1] \times [0,1]$, divided into $N^2$ equal squares; the integration steps $h_r$ and $h_z$ are equal to $\frac{1}{N}$.

The stability and convergence of the computational algorithm are studied in the formulation of thickening grids. The grinding of the spatial step entails a significant decrease in the time step with saving the Courant number, unlike the previously developed algorithm for the spherical grain of the catalyst [15,16]. However, conducted calculations with a step reduction in space ($h = 0.1$, $h = 0.01$, $h = 0.001$) while preserving the Courant number showed the convergence of the algorithm.

### 4.2. The Algorithm Implementation

The difference scheme formed the basis of the software implemented in C++ using MPI parallel computing technology.

A block scheme was used to decompose the integration area. It was divided into several equal squares, which made it possible to reduce data exchange to the exchange of boundary conditions. Calculations for each subarea were carried out on a separate computing node for the spatial grid $10 \times 10$. A step-by-step calculation of kinetic equations by the Radau method, calculation of diffusion flows, calculation of transfer processes and calculation of heat transfer by the finite element method is performed after initialization of functions and initial data at each node. The parallel algorithm was analyzed to efficiency (Table 1).

**Table 1.** The acceleration and efficiency of the parallel algorithm.

|  | Acceleration | Efficiency |
|---|---|---|
| 1 processor | 1.0000 | 1.0000 |
| 2 processors | 1.9584 | 0.9792 |
| 4 processors | 3.9177 | 0.9794 |
| 8 processors | 7.7658 | 0.9707 |
| 12 processors | 11.5032 | 0.9586 |

The analysis showed rather stable indicators for several processors, which confirms the validity of using multiprocessor computing for this task for increased time costs.

## 5. Results

### 5.1. Description of the Experiment

The following values of the model parameters were selected, corresponding to the actual conditions of oxidative regeneration for the computational experiment:

$$\rho_C = 1.8 \text{ ton/m}^3, \gamma_k = 0.7 \text{ ton/m}^3, \varepsilon_k = 0.5, \tau_k = 4.8 \text{ s}, c_0 = 15 \text{ mole/m}^3, \quad (16)$$

$$R_C^0 = 10^{-8} \text{ m}, \ T_{op} = 793 \text{ K}, \ M_C = 12 \text{ g/mole}, \ \theta_1^0 = 0.12, \ \theta_2^0 = 0, \ y_1^0 = 0.05, \quad (17)$$

$$\alpha = 11.5 \text{ Wt/(m} \cdot \text{s)}, \ \beta = 0.0115 \text{ m/s}, \ N = 10. \quad (18)$$

The constants of the velocities at a temperature of 720 K and the activation energy for the stages (1) are given in the works [17,18].

The thermal conductivity and heat capacity are approximated by polynomials of the second degree based on tabular data for aluminum oxide. The values of the heat capacity as a function of temperature for the catalyst material (aluminum oxide) are given in [26]. The experimental data given in [27] are taken to approximate the thermal conductivity coefficient of the catalyst material. The effective coefficient of thermal conductivity is calculated taking into account the inclusions of coke in the catalyst; data on the characteristics of coke are given in the book [28].

Computational experiments are given for different values of the length and radius of whole and hollow catalyst grains at an initial grain temperature of 273 K and a gas temperature of 793 K. The initial mass fraction of coke is 3% of the catalyst grain weight; the oxygen concentration in the reaction mixture is 5% (vol.).

Below are the pictures of the distribution of coke and of the gas-phase substances along the longitudinal section of the whole cylindrical grain of the catalyst. Visualization is performed in the Paraview package.

### 5.2. Distribution of Coke by Catalyst Grain

The main characteristic of the process of burning coke sediments is a decrease in the mass fraction of coke in the catalyst grain.

Figure 1 captures the change in the radius-averaged mass fraction of coke over time. The distribution of coke over the catalyst grain at 24 and 32 min of the process is shown in Figures 2 and 3.

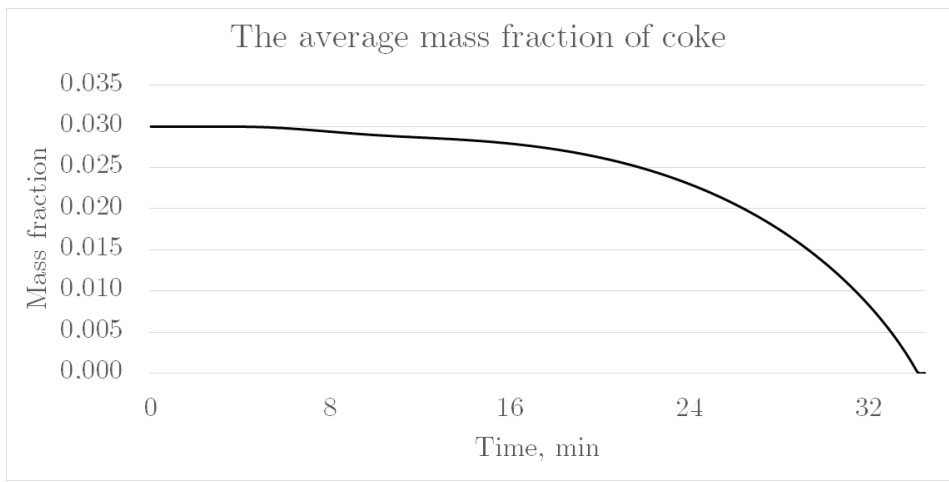

**Figure 1.** Dynamics of the average mass fraction of coke in the pores of the catalyst grain.

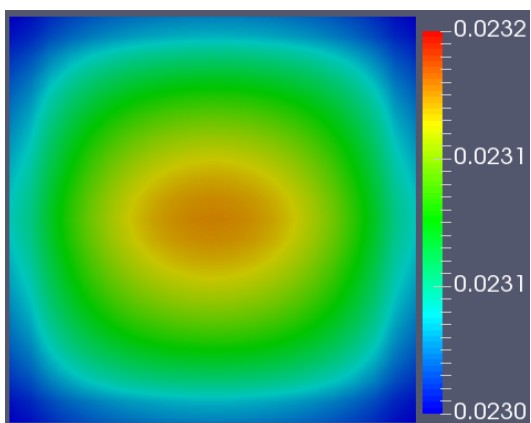

**Figure 2.** Coke distribution patterns at 24 min.

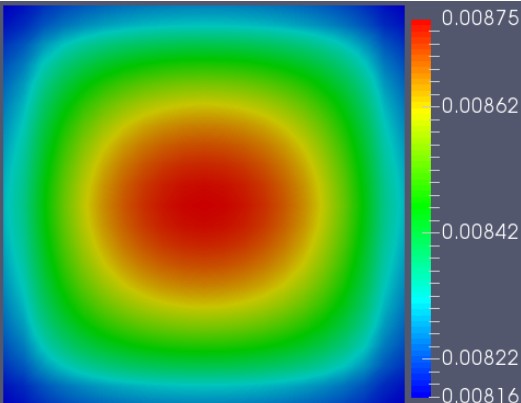

**Figure 3.** Coke distribution patterns at 32 min.

It can be seen from Figures 2 and 3 that coke burns out from the grain boundary to its center, which corresponds to a similar distribution of oxygen due to its entry through the boundary into the pores of the grain.

### 5.3. Distribution of the Gas-Phase Substances of the Reaction

The pictures of the distribution of the gas-phase substances along the longitudinal axial section of a whole cylindrical grain of the catalyst at various points in time are presented in Figures 4–7.

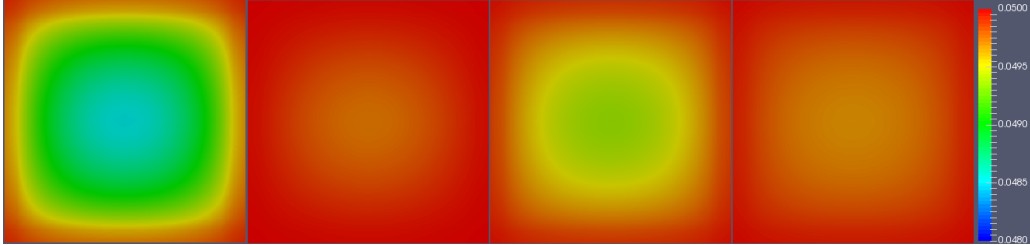

**Figure 4.** Oxygen distribution patterns over grain for 8 min, 16 min, 24 min and 32 min.

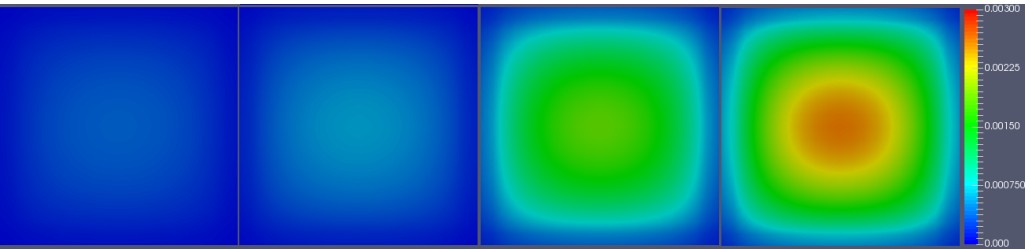

**Figure 5.** Patterns of carbon monoxide distribution over grain for 8 min, 16 min, 24 min and 32 min.

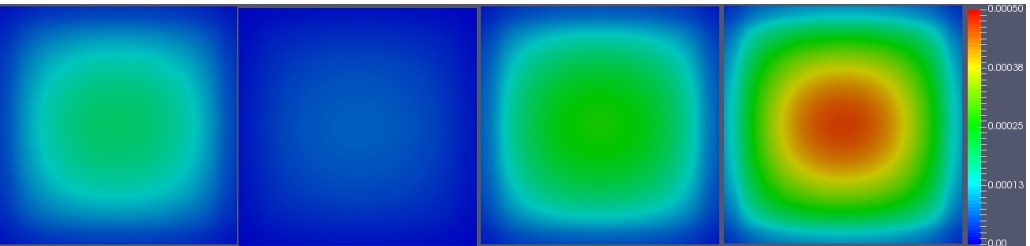

**Figure 6.** Patterns of carbon dioxide distribution over grain for 8 min, 16 min, 24 min and 32 min.

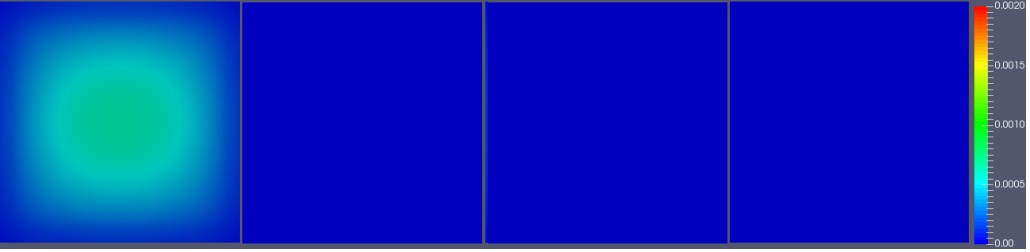

**Figure 7.** Patterns of water distribution over grain for 8 min, 16 min, 24 min and 32 min.

The distribution of the gas-phase substances along the catalyst grain corresponds to the kinetic processes of the (1) scheme.

The book in ref. [17] describes the composition of coke sediments. The following assumptions are made in this article: coke consists of a light-burning part (carbon–hydrogen and carbon–oxygen compounds) and a hard-burning part (clear carbon).

The light-burning part of the coke, consisting of hydrocarbon compounds, was oxidized in 8 min of process. This explains the presence of carbon oxides and water in the grain. The water concentration decreases due to the mass transfer process after the burning out of the light-burning component of coke.

Figures 8–11 show the change over time of the molar fractions of the gas-phase components averaged over the grain radius.

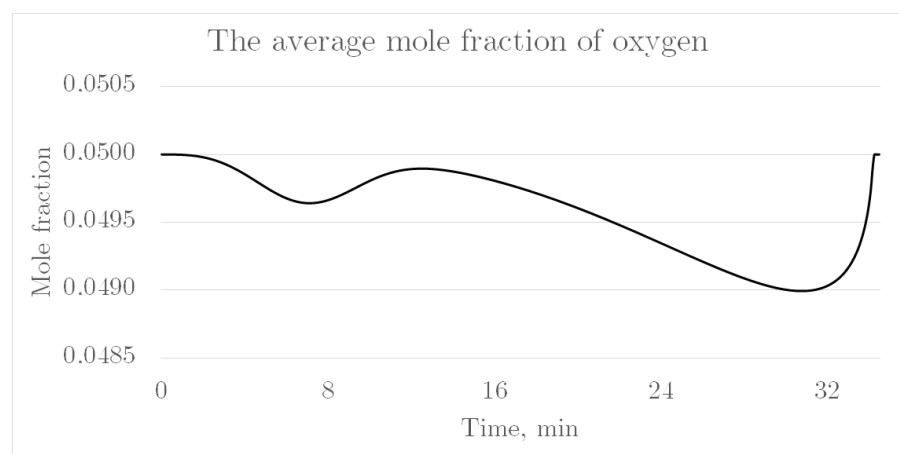

**Figure 8.** Dynamics of the average molar fraction of oxygen in the reaction mixture in the pores of the catalyst grain.

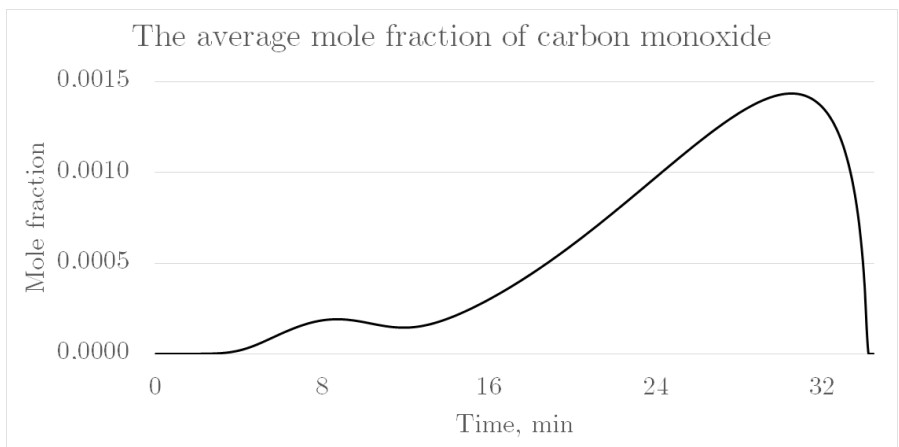

**Figure 9.** Dynamics of the average molar fraction of carbon monoxide in the reaction mixture in the pores of the catalyst grain.

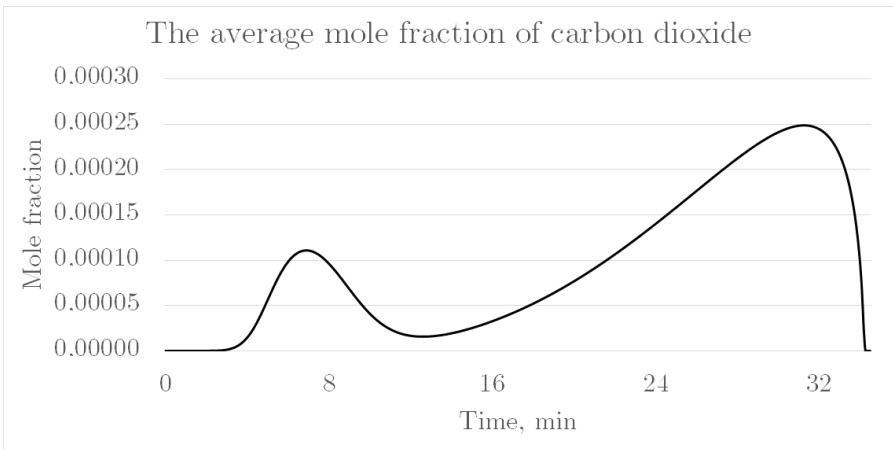

**Figure 10.** Dynamics of the average molar fraction of carbon dioxide in the reaction mixture in the pores of the catalyst grain.

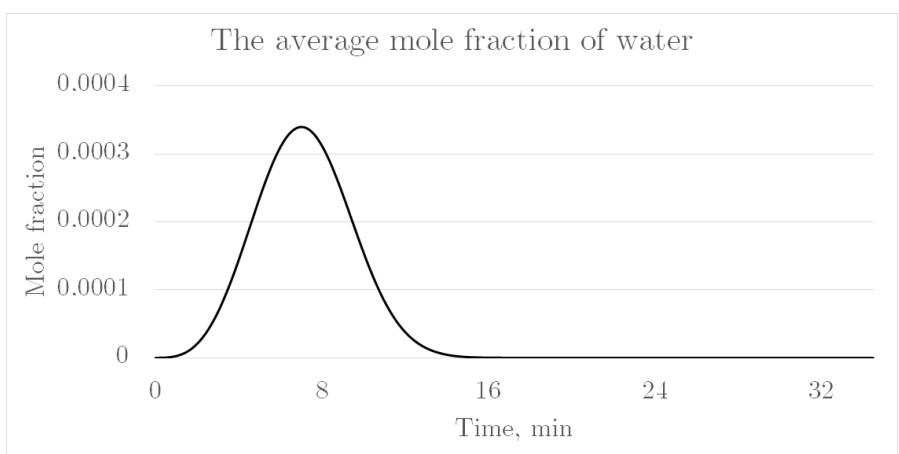

**Figure 11.** Dynamics of the average molar fraction of water in the reaction mixture in the pores of the catalyst grain.

As follows from Figures 8–11, the dynamics of the averaged characteristics of the gas phase correspond to the dynamics of the characteristics in the spherical grain [15]. As can be seen from Figure 1, the mass fraction of coke in the catalyst grain is noticeably reduced at the time of 16 min. At the same time, an increase in the concentration of carbon oxides and a decrease in the concentration of oxygen can be noted. This is due to the beginning of the combustion of the free carbon surface.

*5.4. Catalyst Grain Temperature Dynamics*

Figure 12 shows the dynamics of the radius-averaged catalyst grain temperature.

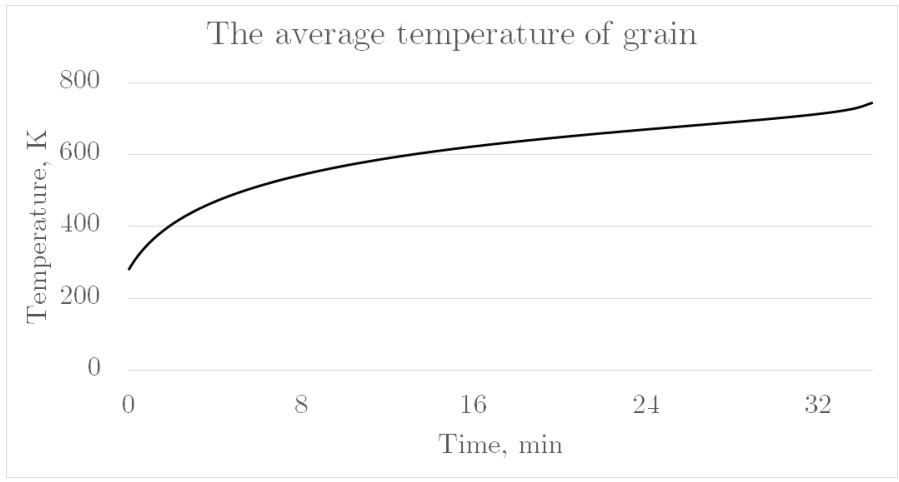

**Figure 12.** Dynamics of the average temperature of the catalyst grain.

The heating of the grain is caused by heat exchange through its boundaries, as well as the course of exothermic reactions.

The conclusion about the uniformity of the temperature distribution over the catalyst grain during the calculations was made. It corresponds to the hypothesis about the isothermicity of the grain [6,7,15]. This will make it possible to avoid taking into account the temperature distribution over the catalyst grain when modeling a cylindrical catalyst layer.

**6. Discussion**

The kinetic model of the reaction proposed in [17], and previously applied by the authors to obtain an oxidative regeneration model for a spherical grain, is used in this article.

It is possible to distinguish similar moments in the patterns of temperature distribution and substances across the grain due to the commonality of chemical processes in spherical

and cylindrical grains. The concentration profiles of reagents and reaction products are qualitatively the same, however, in the cylindrical grain, the concentration fluctuations are stronger, which is due to the modification of the model in terms of the use of nonconstant thermophysical characteristics. Moreover, a sharper concentration gradient is observed in the distribution patterns in the cylindrical grain. This is due to the assumption about the uniformity of the flow of processes in all directions being valid for a spherical grain shape, but it is necessary to take into account the transfer processes not only along the radius but also along the axis of the cylinder for a cylindrical shape.

The temperature distribution coincides qualitatively and quantitatively in both cases: the heating is the most intense in the first minutes due to the large difference in the temperatures of the grain and the gas layer outside. With the passage of time , equilibrium is established and, by the time of complete burnout of coke sediments, the grain temperature is about 753 K. The time of the process also coincides—the burning lasts about 32 min.

## 7. Conclusions

The article develops the mathematical model of the oxidative regeneration of a cylindrical catalyst grain using a diffusion approach—diffusion, Stefan flow, thermal conductivity of the grain and chemical reactions are taken into account.

The model is presented in an axisymmetric formulation. The calculation of the main characteristics was conducted for whole catalyst grains. This corresponds by the boundary conditions: for a whole grain, heat and mass transfer occurs through the cylindrical wall and the ends of the cylinder.

The calculation of the thermal balance of the catalyst grain was carried out using the effective coefficients of heat capacity and thermal conductivity of the catalyst grain, taking into account its porosity. The dependence of these coefficients on temperature is restored on the basis of experimental data.

The efficient computational algorithm is constructed, taking into account the very different times of diffusion processes and chemical reactions. Splitting by physical processes with sequential calculation of chemical reactions, diffusion and thermal conductivity were used to reduce the complexity of the algorithm. The resulting difference scheme formed the basis of the code that allows calculations of the main characteristics of the process.

Technological parameters corresponding to the actual process of burning coke sediments are used for the computational experiment.

The result of the program is the distribution patterns of substances and temperature by the catalyst grain.

**Author Contributions:** Conceptualization, I.M.G.; methodology, O.S.Y.; software, E.E.P.; validation, A.N.Z.; formal analysis, L.A.S. All authors have read and agreed to the published version of the manuscript.

**Funding:** This research was partially funded by the state task of the Institute of Petrochemistry and Catalysis of the Russian Academy of Sciences (theme No. FMRS-2022-0078) and the Boreskov Institute of Catalysis (project AAAA-A21-121011390010-7).

**Data Availability Statement:** Not applicable.

**Conflicts of Interest:** The authors declare no conflict of interest. The funders had no role in the design of the study; in the collection, analyses, or interpretation of data; in the writing of the manuscript; or in the decision to publish the results.

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
