# Peer review of "Computer Simulation of Coke Sediments Burning from the Whole Cylindrical Catalyst Grain"

_mathematics, doi:10.3390/math11030669_

Round 1

Reviewer 1 Report

The paper deals with the development of the mathematical model of oxidative regeneration of the cylindrical catalyst grain. Model described by a stiff system of differential equations and divided in order to solve chemical problems, then the diffusion and thermal conductivity. The major objections are given below:

1.       Previously, the authors published a mathematical model for burning coke sediments from a spherical grain of the catalyst. Please specify the main similarities and/or dissimilarities with result obtained in this study.

2.       Explain W6 and W7 due to these reactions are reverse, we have W-6 and W-7. Also, explain the connection with density and the average radius.

3.       How did you measure the activation energies?

4.       How do you get equations (14)? Please provide adequate Supplementary Material.

5.       Define N

6.       Use the identical unit (K or C) for temperature through text.

7.       What is catalyst in this process? It is not clearly understand.

8.       Discussion is not adequate. The current one corresponds to the conclusion.

9.       Conclusion is missing.

Author Response

The author's team expresses gratitude for the careful reading and comments. The answers and comments are contained in the attachment.

Reviewer 2 Report

The manuscript entitled “Computer simulation of coke sediments burning from the whole cylindrical catalyst grain”. It deals with the development of the mathematical model of oxidative regeneration of the cylindrical catalyst grain.

The work is interesting and precedent, because the use of catalysts significantly reduces the decomposition temperature of substances in contact with the active sites present on their surfaces. The major problem in substrate decomposition to produce desired products is bed clogging due to carbon deposition, requiring periodic regeneration of the catalyst. In this way, activation, reaction and regeneration cycles are often used in this type of process.

Therefore, I consider its publication in the Journal Mathematics acceptable after review, taking into account the following points.

In the introduction, the manuscript reports that previously, the authors obtained a mathematical model to burn coke sediments from a spherical catalyst grain [6,7], developed based on the well-known model [8,9]. So, it would be suggestive to highlight in the manuscript the relevance of this new study, highlight the novelty in comparison to previous works.

Most of the most recent references used in this work are authored by those involved in it. What makes self-citation. Therefore, it is strongly recommended that you add references to other authors related to this work.

Were the results presented in Figures 1-12 obtained by any specific technique for characterizing catalysts during oxidative regeneration? If so, mention in the manuscript.

Section 6. Discussion looks more like the conclusion of the work. If not, I suggest that you include a conclusion section in the manuscript.

Author Response

(The authors gave the same response as above.)

Reviewer 3 Report

The article titled: "Computer simulation of coke sediments burning from the whole cylindrical catalyst grain" tackles an important issue in chemical engineering and mechanics. In addition, the article covers the area of computational and applied mathematics, and therefore is suitable for Mathematics.
Article includes the development of the mathematical model of oxidative regeneration of the cylindrical catalyst grain. The results in the study are achieved via experimental study.

Comments:
1) In my opinion, the novelty of the research is debatable. I think the Authors should indicate the novelty and purpose of research.
2) The state of the art should be presented clearly with a description of a particular publication’s contribution to its development. The collective citations in: line 29, line 54, line 188, are not acceptable in a scientific journal.
3) Figures 1, 8, 9, 10 and 11 should be corrected. The values on the y-axis are incorrectly represented.
4) Equation number on page 2 omitted.
5)  The article contains adequate and appropriately selected 18 literature items. The presented literature includes only three current literature (from the last three years). In my opinion, the article should be enriched with a more extensive review of the current scientific literature.

The paper is well prepared and written in good English.
In opinion of the reviewer the article needs minor corretion/complete with data and provide a specific purpose aim.

Author Response

(The authors gave the same response as above.)

Reviewer 4 Report

1.      A clearer explanation of the kinetic scheme presented in 1 is necessary, considering that the proposed reactions and the proposed rates Wi are not the only possibilities, but that there are other proposed mechanisms.

2.      This type of combustion processes are usually easily modeled (of coke) using the unreacted core model or the progressive conversion model, what is the advantage of the proposed model?

3.      Equation 2 can be used in case all the reactions shown in scheme 1 are considered to present the same rate increase with temperature, which is clearly not the case, is the model still adequate in such a situation? explain.

4.      The present model may not take porosity into account, which is very important in diffusion reactions. Explain

5.      In order to better analyze the results of the model, it is suggested that the simulation be run again at a different temperature than 520 °C, and compare the results in figure 1 and 2, I suggest 550 and 600 for example, of this In this way, it will be possible to evaluate how the scietics changes with this parameter and obtain better conclusions.

6.      Likewise, I suggest that the model at 520 °C be run again with two other initial particle sizes. Only the analysis of the results shown in the form of turns 1 and 2 would be interesting.

Author Response

(The authors gave the same response as above.)

Round 2

Reviewer 1 Report

Dear Authors, 

Thank you for your corrections. The next time, all additions and corrections in the main text should be put in color for a quick review.

Best regards, 

Reviewer 2 Report

I recommend the publication of the manuscript

Reviewer 4 Report

The authors have responded satisfactorily to the previous observations